# Rationally patterned electrode of direct-current triboelectric nanogenerators for ultrahigh effective surface charge density

Zhihao Zhao [1,2,5], Yejing Dai [2,5], Di Liu [1,3,5], Linglin Zhou[1,3], Shaoxin Li[1,3], Zhong Lin Wang [1,3,4✉] & Jie Wang [1,3✉]

As a new-era of energy harvesting technology, the enhancement of triboelectric charge density of triboelectric nanogenerator (TENG) is always crucial for its large-scale application on Internet of Things (IoTs) and artificial intelligence (AI). Here, a microstructure-designed direct-current TENG (MDC-TENG) with rationally patterned electrode structure is presented to enhance its effective surface charge density by increasing the efficiency of contact electrification. Thus, the MDC-TENG achieves a record high charge density of ~5.4 mC m$^{-2}$, which is over 2-fold the state-of-art of AC-TENGs and over 10-fold compared to previous DC-TENGs. The MDC-TENG realizes both the miniaturized device and high output performance. Meanwhile, its effective charge density can be further improved as the device size increases. Our work not only provides a miniaturization strategy of TENG for the application in IoTs and AI as energy supply or self-powered sensor, but also presents a paradigm shift for large-scale energy harvesting by TENGs.

[1] Beijing Institute of Nanoenergy and Nanosystems, Chinese Academy of Sciences, Beijing 100083, China. [2] School of Materials, Sun Yat-sen University, Guangzhou 510275, China. [3] College of Nanoscience and Technology, University of Chinese Academy of Sciences, Beijing 100049, China. [4] School of Materials Science and Engineering, Georgia Institute of Technology, Atlanta, GA 30332, USA. [5] These authors contributed equally: Zhihao Zhao, Yejing Dai, Di Liu. ✉email: zhong.wang@mse.gatech.edu; wangjie@binn.cas.cn

As a common phenomenon, the contact electrification (CE) has been known for a long time[1]. Based on the fundamental physical mechanism of CE, the triboelectric nanogenerator (TENG) was first invented by Wang and colleagues[2], which provides a strategy for converting randomly distributed and irregular mechanical energy into electric energy[3–5]. Various TENGs have been intensively conducted in two categories: (i) coupling CE with electrostatic induction, the TENG gives an alternating current (AC-TENG)[6–8]; (ii) coupling CE with electrostatic breakdown, the TENG generates a direct current (DC-TENG)[9,10]. These different types of TENGs provide effective techniques for harvesting distributed mechanical energy, wave energy, and biomechanical energy, and show a great potential in application of Internet of Things, implantable medical devices, and artificial intelligence as micro/nano energy or self-powered sensors[11–16].

As an energy harvester, how to improve the output performance of TENG, which is determined by its surface charge density quadratically[17], is a crucial problem for its commercial applications. The limitation factors of effective surface charge density of AC-TENG ($\sigma_{\text{AC-TENG}}$), which can be effectively converted into electric power to drive an external load, can be described as[18]:

$$\sigma_{\text{AC-TENG}} = \min\left(\sigma_{\text{triboelectrification}}, \sigma_{\text{r,air breakdown}}, \sigma_{\text{dielectric breakdown}}\right) \quad (1)$$

where $\sigma_{\text{triboelectrification}}$ is the triboelectrification charge density, $\sigma_{\text{r, air breakdown}}$ is the remaining surface charge density after the air breakdown between two friction surfaces, and $\sigma_{\text{dielectric breakdown}}$ is the maximum charge density that the dielectric can store. The charge density of TENGs can be increased with the enhancement of $\sigma_{\text{triboelectrification}}$ by materials optimization and structure design[19–21]. However, with rising charge density on dielectric surface, the air breakdown will occur between two friction surfaces and part of charges will be released, resulting in the limitation of $\sigma_{\text{r, air breakdown}}$. High-vacuum environment can avoid the air breakdown and thus significantly improve the charge density of TENG up to ~1 mC m$^{-2}$[18]. Ultrathin friction dielectric film is another strategy to elevate the threshold of $\sigma_{\text{r, air breakdown}}$[22]. Furthermore, taking advantage of external circuit optimization to break through the limitation of $\sigma_{\text{triboelectrification}}$, e.g., the charge pumping[23,24] and charge excitation[25,26], the charge density reaches to a milestone of 2.38 mC m$^{-2}$[26]. However, the effective surface charge density of TENG is still limited by the dielectric breakdown of friction dielectric layer ($\sigma_{\text{dielectric breakdown}}$).

As a new type of TENG, the DC-TENG can directly power electronic devices without the auxiliary rectifier circuits and energy storage units[9]. The working mechanism of DC-TENG is based on the triboelectrification effect and the electrostatic breakdown between the friction surface and the charge-collecting electrode (CCE; detailed in Supplementary Note 1), which is free from the limitation of $\sigma_{\text{dielectric breakdown}}$. Therefore, the limitation of its effective surface charge density ($\sigma_{\text{DC-TENG}}$) can be described as:

$$\sigma_{\text{DC-TENG}} = \min\left(\sigma_{\text{triboelectrification}}, \sigma_{\text{c, electrostatic breakdown}}\right) \quad (2)$$

where $\sigma_{\text{c, electrostatic breakdown}}$ is the collected charges from electrostatic breakdown (generally air breakdown), which can be improved by the enhanced thermionic emission of electrons or the avalanche breakdown effect[27]. However, the reported maximum value is only 0.64 mC m$^{-2}$ due to the limitation of $\sigma_{\text{triboelectrification}}$[27], which is lack of the accumulation process of triboelectric charges compared with AC-TENG.

Here we provided a strategy to significantly enhance the charge density of DC-TENG by microstructural design with rationally patterned electrode structure, whose limitation factor can be described as follows:

$$\sigma_{\text{DC-TENG}} = k \times \min\left(\sigma_{\text{triboelectrification}}, \sigma_{\text{c, electrostatic breakdown}}\right) \quad (3)$$

where the $k$ is a factor related to the electrode structure. The microstructure-designed DC-TENG (MDC-TENG) realizes the miniaturized sliding block structure and high-output performance at the same time. By tailoring the electrode structure (where $k = 50$), the effective surface charge density of MDC-TENG with the size of 1 cm × 5 cm can be improved to 5.4 mC m$^{-2}$, which is more than two times of existing record for various type of TENGs. Of particular significance is that the charge density of the MDC-TENG can be further improved with a larger size and a higher $k$-value. Except for the high-output performance, its output current is closely related with the motion vector parameters, such as velocity, acceleration, and distance. These excellent performances represent potential applications of the MDC-TENG in mechanical energy harvesting and motion vector sensing. Especially, its advantages of miniaturization and simple external circuit resulted from DC output provide a solution strategy for TENGs to be applied in small electronic device systems or micro-electro-mechanical system (MEMS) as an energy supply resource or self-powered sensor. Moreover, the significantly enhanced charge density for the large-sized TENG also shows huge potential for the large-scale energy-harvesting application.

## Results

**Structural design and working mechanism of MDC-TENG.** The structure of MDC-TENG with rationally patterned electrode is presented in Fig. 1a–c, which possesses multiple fine friction electrodes (FEs, material: copper wire) and interlaced CCEs (material: stainless-steel wire). All of the individual FEs keep a tiny distance with the adjacent CCEs and there is a very narrow gap existing between the CCEs and the friction layer (polytetrafluoroethene, PTFE), as shown in Fig. 1b. It can be seen from the scanning electron microscopy (SEM) image of the MDC-TENG sample (Fig. 1c) that each FE is about 250 μm in width and the CCE (~100 μm in width) is located between two FEs. The distance between two adjacent FEs is about 1000 μm, so is the distance between two adjacent CCEs. All of the FEs and CCEs are embedded into the acrylic substrate. When the FE rubs with the friction layer, electrons transfer from FE to friction layer due to triboelectrification effect and then a DC is produced due to the air breakdown between the CCE and the charged friction layer (Fig. 1d, detailed mechanism of DC-TENG is shown in Supplementary Fig. 1 and Note 1).

The reason why the FE width can be miniaturized to microscale dimension in this work can be explained by Fig. 1e. Prior to the atomic-scale contact of two materials, their respective electron cloud keeps separate without overlap (Fig. 1e1). When the slide block slides forward and contacts with the friction layer, the electron transition from M1 to D1 occurs (M and D represents the position on the surface of metal FE and dielectric friction layer, respectively) and the energy potential barrier difference between two materials becomes lower (Fig. 1e2), which is called Wang transition model, and has been confirmed by the atomic force Kevin probe microscopy recently[4,28,29]. Thus, few electrons would transit from M2 to D1, owing to their lower potential difference when M2 overlaps D1 (Fig. 1e3). This assumption has been confirmed by the output performance of DC-TENGs with different widths of FE, as shown in Supplementary Fig. 2. It can be seen that, for the same length and sliding

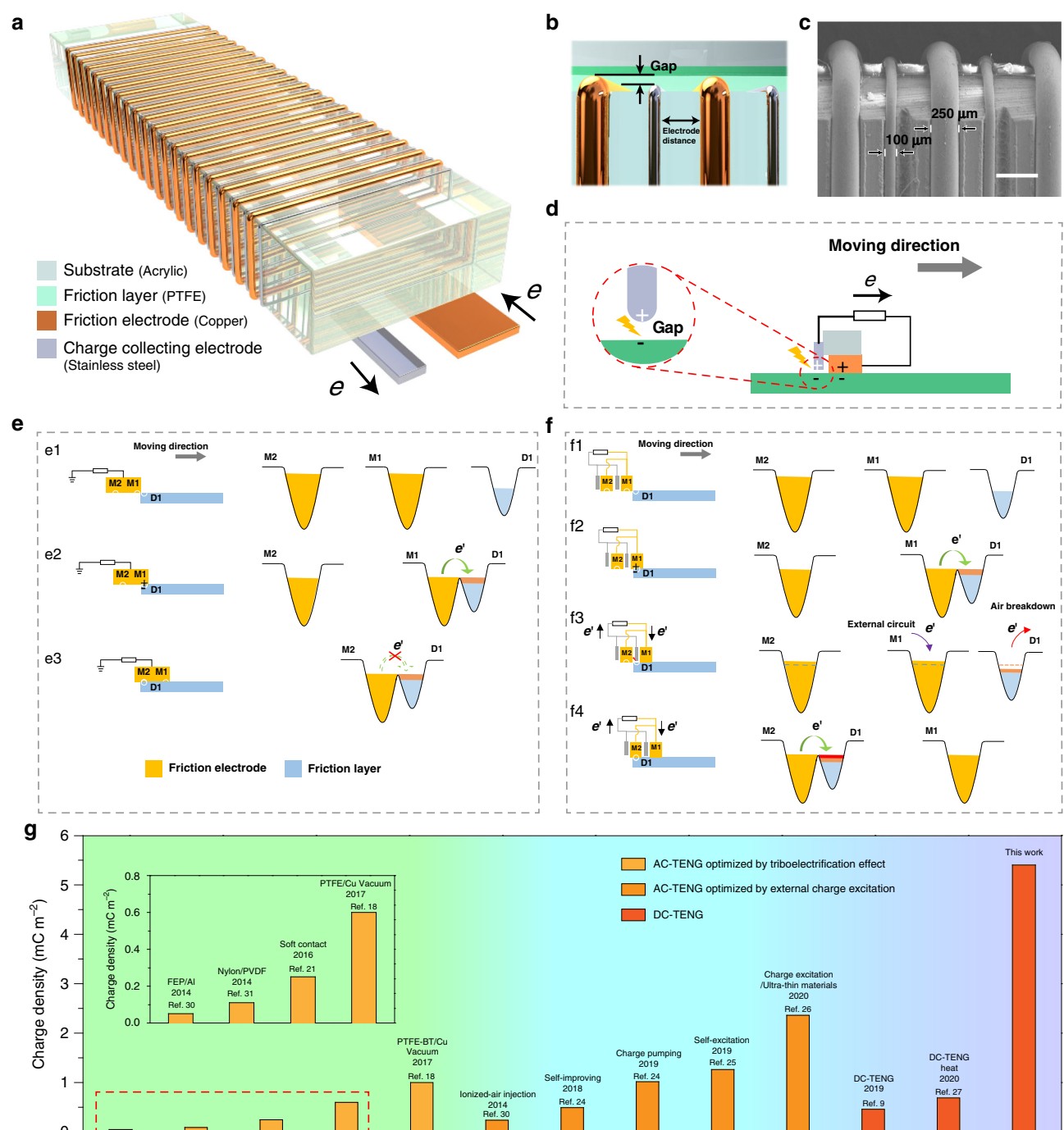

**Fig. 1 Structural design and working mechanism of MDC-TENG. a**, **b** Structural schematic and **c** SEM image of MDC-TENG (scale bar: 500 μm). **d** Schematic diagram of DC-TENG working mechanism. The schematic diagram of electrons transition process **e** without and **f** with microstructure optimization. **g** Comparison of the charge density of MDC-TENG with different type TENGs.

distance of FE, the output charges and short-circuit currents of all the DC-TENGs are not significantly different with the FE width decreasing from 10 to 0.25 mm. Namely, the FE width has little effect on the $\sigma_{\text{triboelectrification}}$ in Eq. (2) due to the high triboelectrification efficiency of the sliding TENG[4].

To improve the utilization rate of FE and efficiency of TENG, we provide a method to replace a single large-scale FE by using patterned multiple micrometer-size FEs, interlaced with micrometer-size CCEs in two adjacent FEs (M1 and M2). The working mechanism can be described in Fig. 1f. The initial stage (Fig. 1f1) and CE process between M1 and D1 (Fig. 1f2) are

similar to those in Fig. 1e1, e2, respectively. However, with the slider moving forward, electrostatic breakdown will occur between CCE and D1 (fundamental mechanism is shown in Supplementary Note 1), and electrons will transfer from D1 to CCE, and then to M1/M2 via external circuit, causing the energy potential barrier difference to form again between M2 and D1. Thus, the CE process will take place again between M2 and D1 due to the Wang transition model and the following CCE can collect the generated charges. Consequently, combining electrode microstructural design with electrostatic breakdown effect, electron transition of CE occurs twice within the same width of

electrode compared with Fig. 1e. Based on the above mechanism, an MDC-TENG with more elaborately designed patterned electrodes is prepared and an ultrahigh effective surface charge density of ~5.4 mC m$^{-2}$ is achieved with 50 FEs.

The development of effective surface charge density in AC-TENG and DC-TENG is shown in Fig. 1g[9,18,21–27,30,31]. After the unremitting efforts of researchers in recent years, many approaches, e.g., high-vacuum[18], ion injection[30], charge pumping[23,24], and charge excitation[25,26], were carried out (Supplementary Note 2). The limitation factors in Eq. (1) were broken gradually and charge density of AC-TENG was successfully improved from <0.05 mC m$^{-2}$ to over 2.3 mC m$^{-2}$[18,21,23–26,30,31]. As for DC-TENG, its typical charge density of 0.4 mC m$^{-2}$ was reported in 2019, which is relatively low compared to the AC-TENG in the same period[9]. However, in this work, the charge density of 5.4 mC m$^{-2}$ is not only over tenfold higher than that of DC-TENG reported in 2019, but also over two times than that of the state-of-the-art of various type of TENGs (Fig. 1g).

**Comparison between sliding AC-TENG and MDC-TENG.** To confirm the superiority of MDC-TENG over the traditional AC-TENG, the output performance of MDC-TENG and AC-TENG was carried out and shown in Fig. 2. Working principle of the sliding AC-TENG is presented in Fig. 2a. The slider contacts with the friction layer, generating opposite charges on the surface of electrode and friction layer due to the CE effect, respectively. When the slider continues to move forward, the relative displacement between two electrodes makes the potential difference between them and the electrons flow in the external circuit to balance this potential difference (Fig. 2a). Figure 2b–e show the effective surface charges, charge density, short current, and current density of sliding AC-TENG (friction layer: PTFE) with different electrode lengths and sliding distances (electrode length: $x$ mm, sliding distance: $y$ mm, $x = y$ in this test, electrode width: 10 mm). With the $x$ and $y$ increasing, output charges increase from 6 to 68 nC, accompanied with charge density ~0.12 mC m$^{-2}$ (Fig. 2b, c). The short current $I_{sc}$ increases fast with the device size and sliding distance increasing, but the current density decreases from 510 to 170 μA m$^{-2}$ (Fig. 2d, e).

The schematic of MDC-TENG is shown in Fig. 2f, where one MDC-TENG unit includes one FE and one CCE, electrode length (the distance from the first FE to the last CCE): $x$ mm, sliding distance: $y$ mm, MDC-TENG unit: $n$, $x = y = n$ in this test, electrode width: 10 mm (the photograph of MDC-TENG with 20 units, shown in Supplementary Fig. 3). Figure 2g–j show the effective surface charges, charge density, short current, and current density of MDC-TENG (friction layer: PTFE) with different electrode lengths, sliding distances, and different numbers of MDC-TENG unit. With the $x$, $y$, and $n$ increasing, charges of MDC-TENGs increase from 0.025 to 2.6 μC (Fig. 2g), which are much larger than that of AC-TENG at the same dimension (Fig. 2b). More importantly, charge density of MDC-TENGs rises from 0.5 to 5.2 mC m$^{-2}$ (Fig. 2h), which are nearly 40-fold larger than those of AC-TENGs under the same

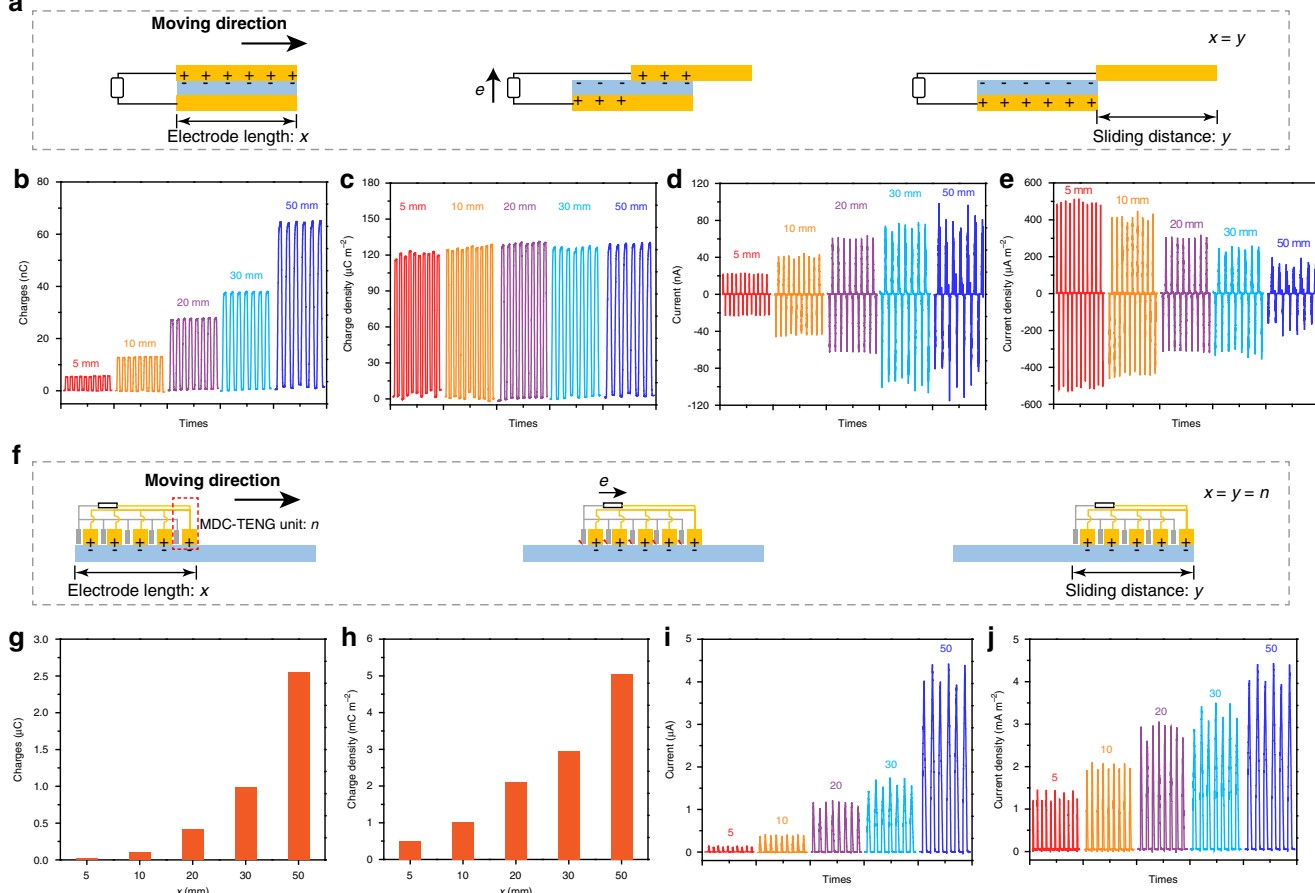

**Fig. 2 Comparison between sliding AC-TENG and MDC-TENG. a** Working mechanism of sliding AC-TENG. The **b** output charges, **c** charge density, **d** short current, and **e** current density of sliding AC-TENG (friction layer: PTFE) with different electrode length and sliding distances. **f** Working mechanism of sliding MDC-TENG. The **g** output charges, **h** charge density, **i** short-circuit current, and **j** current density of MDC-TENG (friction layer: PTFE) with different electrode length, sliding distances, and different numbers of MDC-TENG units.

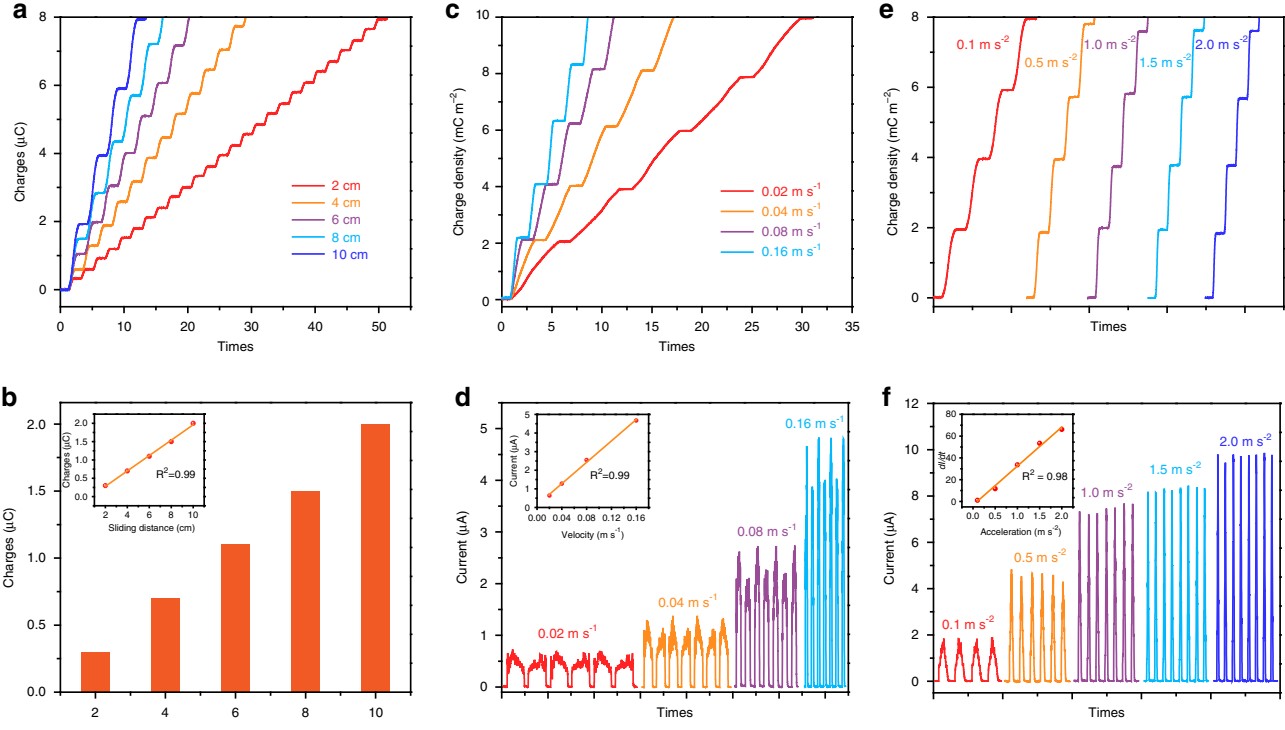

**Fig. 3 Performance of MDC-TENG under different vector parameters. a, b** The output charges under different sliding distances (the inset is the relationship between output charge and sliding distance). **c** The charge density and **d** short-circuit current of MDC-TENG with various velocities (the inset is the relationship between $I_{sc}$ and velocity). **e** The charge density and **f** short-circuit current of MDC-TENG with various accelerations (the inset is the relationship between $dI/dt$ and acceleration).

condition. The high charge density of MDC-TENGs with $n = 50$ is due to the multiple electron transition through the repeated triboelectrification and discharge processes (Fig. 1f). Meanwhile, $I_{sc}$ increases fast with the increase of $x$, $y$, and $n$ (Fig. 2i), accompanied with the enhanced current density (Fig. 2j). In general, the traditional AC-TENG improves the contact efficiency by reducing the contact area to achieve a high charge density[4,21], and thus the charge density and current density (Fig. 2c, e) decrease with the size of AC-TENG device increasing. However, the enhanced charge density and current density with the enlarged size of MDC-TENG addresses this performance attenuation of AC-TENG, which benefits the application in large-scale energy-harvesting system for TENGs.

**Performance of MDC-TENG under various motion parameters.** As a vector motion device, the output performance of sliding mode MDC-TENG (with 20 MDC-TENG units, shown in Supplementary Fig. 3) under various vector motion parameters is shown in Fig. 3. As the slider slides back and forth on the PTFE surface within different distances, the output charge curve shows a stepped-like shape. When the sliding distance is 2 cm, the average charges of MDC-TENG is about 0.35 μC (Fig. 3a). The output charge at each movement process increases with the extension of sliding distance (Fig. 3b), accompanied with the increase of $I_{sc}$ (Supplementary Fig. 4). The output charge of MDC-TENG is proportional to the sliding distance with a high linearity of ~0.99 (the inset in Fig. 3b). Within the sliding distance 10 cm, the output characteristic of MDC-TENG at various velocities (uniform motion) is shown in Fig. 3c, d. The average charge density maintains ~2.0 mC m$^{-2}$ during the sliding velocity increasing from 0.02 to 0.16 m s$^{-1}$, but the average $I_{sc}$ rises rapidly from 0.6 to 4.6 μA. The average $I_{sc}$ shows a good linear relationship with velocity (the inset in Fig. 3d). The detailed

relationship between $I_{sc}$ and velocity is explained in the Supplementary Note 4. Moreover, the output performance of MDC-TENG under different accelerations is shown in Fig. 3e, f. The average charge density also maintains ~2.0 mC m$^{-2}$ at different accelerations. Meanwhile, the $I_{sc}$ increases from 1.8 to 9.8 μA with the acceleration of slider increasing from 0.1 to 2.0 m s$^{-2}$, respectively. The $dI/dt$ is proportional to the sliding acceleration with a high linearity ~0.99 (the inset in Fig. 3b), whose relationship is calculated in the Supplementary Note 5. The output characteristic of MDC-TENG shows a good correlation with the vector motion parameters (e.g., distance, velocity, and acceleration), which is the basis of motion vector sensor. In addition, the size of MDC-TENG can be further miniaturized, whereas the high-output performance ensures the strength and anti-interference of sensing signals[32]. Thus, the MDC-TENG shows great potential in the applications on MEMS as the motion vector sensor unit.

**Structure optimization and output performance of MDC-TENG.** To further optimize the output performance of MDC-TENG, the effect of the electrode distance between FE and adjacent CCE is studied, as shown in Fig. 4a–d. When the width of FE is 250 μm and CCE is 100 μm, the distance between FE and CCE ranges from 800 to 100 μm (Fig. 4a), accompanied with the whole-length MDC-TENG device decreasing from 10 to 3.5 mm (MDC-TENG unit: 5, width of MDC-TENG: 10 mm). Within the slide distance ~10 cm, the output charges are about 0.5 μC for all MDC-TENGs with various electrode distances. The corresponding $I_{sc}$ ~ 0.67 μA and $V_{oc}$ ~ 33 V are also insusceptible with the decrease of electrode distance, indicating the potential of miniaturization of MDC-TENG. On one hand, charge densities calculated by the friction area (10 cm$^2$) of MDC-TENGs with the same electrode number but the decreasing electrode distance

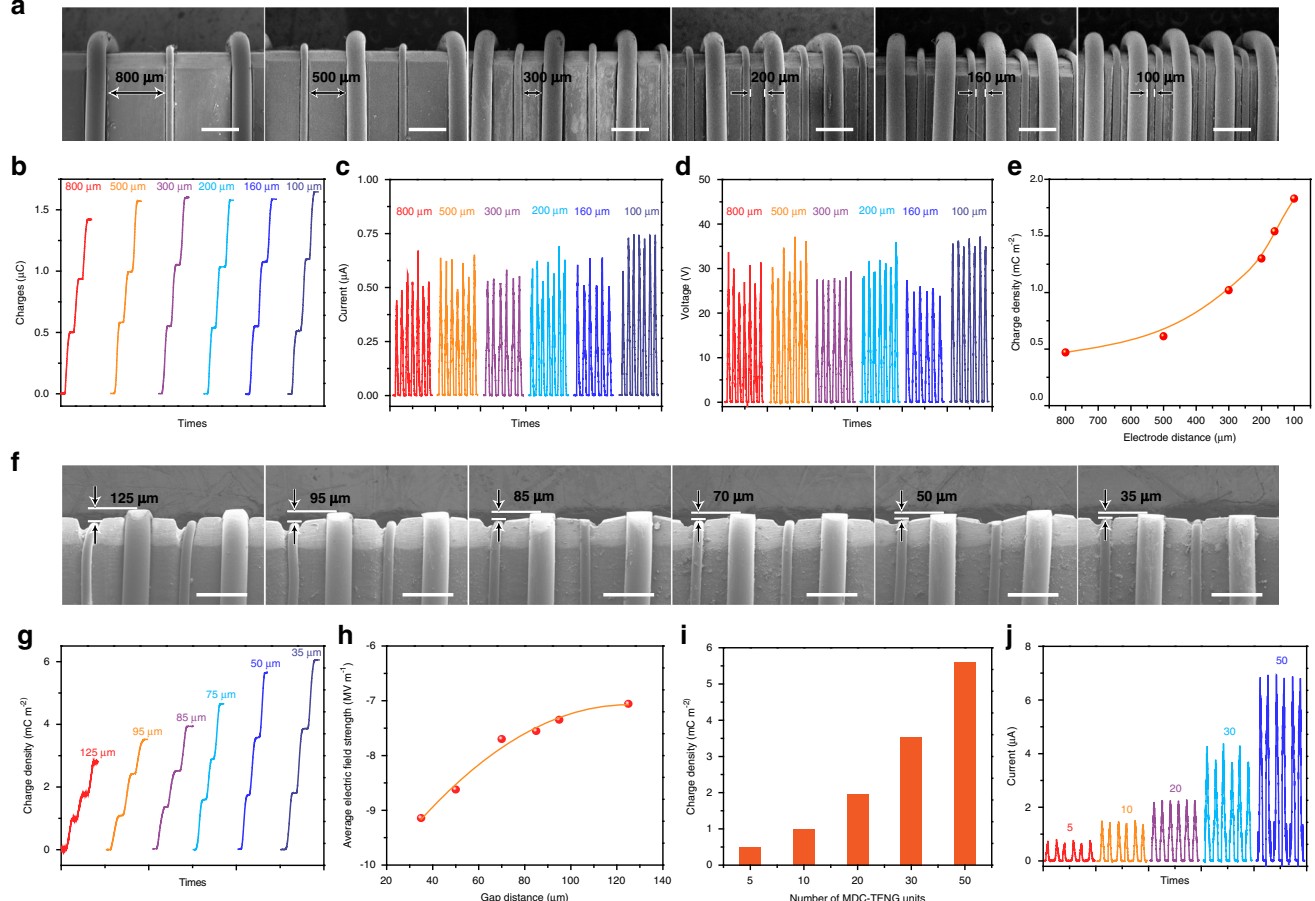

**Fig. 4 Structural optimization and output performance of MDC-TENG. a** The SEM images (scale bar: 500 μm) and corresponded **b** charge density, **c** $I_{sc}$, and **d** $V_{oc}$ of MDC-TENG with different electrode distance (number of MDC-TENG units: 5). **e** Charge density calculated by the friction area and MDC-TENG device area. **f** The SEM images (scale bar: 500 μm) and **g** corresponded charge density of MDC-TENG with different gap distance. **h** The average electric field in the gap between CCE and PTFE surface (simulated by COMSOL software). **i** Charge density and **j** $I_{sc}$ of MDC-TENG with various MDC-TENG units. Test parameters: sliding distance: 10 cm, friction layer: PTFE.

maintains ~0.5 mC m$^{-2}$ (Fig. 4b). On the other hand, the decrease in electrode distance makes the whole area of MDC-TENG device become smaller. It is of particular importance for the miniaturized TENG, because the charge densities calculated by the MDC-TENG device area gradually rise with the electrode distance decreasing (Fig. 4e). In a word, the smaller electrode distance means the larger $k$-value in unit area, resulting in higher output of MDC-TENG device. This is critical for the application of MDC-TENG in small electronic device systems or MEMS as the energy supply resource or sensor unit.

The fundamental mechanism of MDC-TENG is air breakdown in the gap between CCE and friction layer, and excessive gap distance makes it difficult for CCE to collect charges, and there is no DC output in external circuit (Supplementary Fig. 5). Thus, the gap distance between CCE and friction layer is important for the output performance of MDC-TENG by increasing the $\sigma_{c, electrostatic breakdown}$ in Eq. (3). We prepared the MDC-TENGs with the precisely controlled gap distance from 125 to 35 μm. Their corresponding SEM images are shown in Fig. 4f and their charge densities are shown in Fig. 4g. Taking PTFE as friction layer, when the gap is 125 μm within the sliding distance 10 cm, the charge density of MDC-TENG with 20 MDC-TENG units is just 0.49 mC m$^{-2}$. With decreasing the gap distance to 35 μm, the charge density gradually increases to 2.0 mC m$^{-2}$, indicating that the $\sigma_{c, electrostatic breakdown}$ in Eq. (3) increases with the gap distance decreasing. Meanwhile, the $I_{sc}$ (Supplementary Fig. 6) also

significantly increases from 0.5 μA for 125 μm gap to 2.0 μA for 35 μm gap. When the gap decreases to 0, the mechanism of DC-TENG (Supplementary Fig. 1 and Note 1) does not work, because there is no gap for the occurrence of air breakdown and thus the output performance significantly decreases (Supplementary Fig. 7).

To clarify this significantly enhancement output, we analyze the potential distribution in the gap between CCE and PTFE by using COMSOL software at different gap distances. Due to the existence of charges on the PTFE surface (setting value: 120 μC m$^{-2}$ from Fig. 2c), a huge electrostatic field generates between the CCE and the PTFE surface, as shown in Supplementary Fig. 8. The simulated average strength of electrostatic field is shown in Fig. 4h and Supplementary Fig. 9, and the calculation method is shown in the Supplementary Note 6. With the decrease of gap, the electric field increases sharply and reaches to 9.1 MV m$^{-1}$ for the gap ~35 μm, resulting in easier air breakdown and more complete electron release process from PTFE surface to CCE. Thus, the MDC-TENG with smaller gap distance will show larger $\sigma_{c, electrostatic breakdown}$ and more effective charge density.

To further improve the output performance of MDC-TENG, the structure factor $k$ in Eq. (3), which is related to the number of MDC-TENG units, is introduced into the microstructural design of MDC-TENG. Using the PTFE as the friction layer, the charge density and $I_{sc}$ increase linearly with the number of MDC-TENG

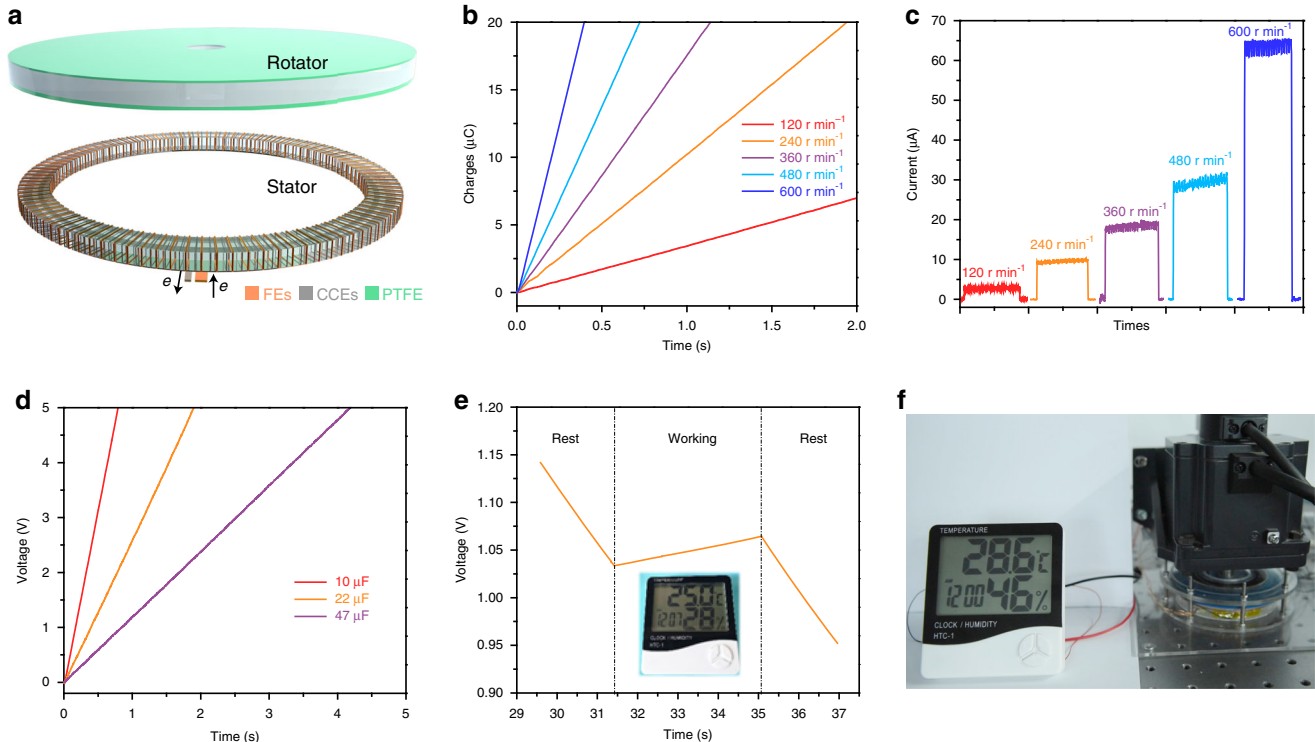

**Fig. 5 Application of MDC-TENG to drive electronic device. a** Structural schematic of rotary MDC-TENG device. **b** Output charges and **c** $I_{sc}$ of rotary MDC-TENG at various rotation rates. **d** Charging curves of 10, 22, and 47 μF commercial capacitors charged by rotary MDC-TENG at 600 r min⁻¹. **e** Charging curve of the 660 μF commercial capacitor with driving thermo-hygrometer simultaneously by rotary MDC-TENG at 600 r min⁻¹. **f** Photograph of driving thermo-hygrometer directly by rotary MDC-TENG at 600 r min⁻¹.

unit under the sliding distance of 10 cm (Fig. 4i, j), so the $k$-value can be simply considered to be equal with the unit number $n$. The charge density of MDC-TENG with $k = 5$ is ~0.5 mC m⁻², but increases to 5.4 mC m⁻² with $k = 50$, which is 50-fold that of the single electrostatic breakdown device (the charge density ~0.11 mC m⁻², as shown in Supplementary Fig. 2). The same trend is also observed in open-circuit voltage $V_{oc}$ curves of MDC-TENG device (Supplementary Fig. 10). This phenomenon is consistent with the previous analysis: the increased efficiency of CE enhances the effective surface charge density, resulting in the number of electron transfers from PTFE to CCE increasing with the adding number of MDC-TENG units within a certain sliding distance.

Except for PTFE, some common organic films, such as Kapton (Polyimide), PVDF (Poly(vinylidene fluoride)), and PPS (Polyphenylene sulfide), were utilized as the friction layers to obtain the output performance of MDC-TENG. As shown in Supplementary Fig. 11, it can been seen that the different friction layers show various output performance, indicating the different ability to produce charges. Within the sliding distance of 10 cm (frication area: 10 cm²), the charge density of MDC-TENG (MDC-TENG units: 20) can be achieved to 1.30, 0.67, and 0.33 mC m⁻² for Kapton, PVDF, and PPS films, respectively, which are lower than that of PTFE.

Based on the above discussion, a record high of charge density of TENG is achieved by microstructural design with the rationally patterned electrodes and the output performance can be further enhanced by the optimization of microstructure, e.g., electrode distance, gap distance, and electrode quantity.

**Application of MDC-TENG for driving electronic device**. As a DC nanogenerator, MDC-TENG could directly drive electronic devices or charge capacitor without bridge rectifier. To achieve

continuous DC output, a rotary mode MDC-TENG was prepared. Its schematic diagram (stator: MDC-TENG) is shown in Fig. 5a. PTFE film is attached on the rotator surface. The stator is a rotary type MDC-TENG device, whose structure is similar to the sliding mode MDC-TENG (Fig. 1a) with interlaced CCEs and FEs. Figure 5b shows the output charges of the rotary MDC-TENG at different rotation rates. With the rotation rate rising, output charge reaches to 20 μC within a short time. For example, it only takes 0.3 s to output 20 μC charges at the rotation rate of 600 r min⁻¹. When the rotary MDC-TENG works stably, the output current is about 65 μA at 600 r min⁻¹ (Fig. 5c), accompanied with a typical DC curve (the crest factor is close to 1). Three commercial LED bulbs (each rated power: 7 W) can also be directly driven with high brightness and no flash due to the high DC output (Supplementary Movie 1).

As an energy harvester device, the harvesting energy of MDC-TENG can also be stored in capacitors or batteries for the subsequent utilization of electronic device. The charging curves of different capacitors charged by the rotary MDC-TENG at 600 r min⁻¹ are presented in Fig. 5d and the detailed circuit is shown in Supplementary Fig. 12. It just takes 0.7, 2.0, and 4.2 s to charge 10, 22, and 47 μF capacitors to 5 V, respectively. In addition, as an energy source, the MDC-TENG can also drive the electronic device and charge energy storage device at the same time. The self-powered system is built by integrating MDC-TENG with the commercial capacitor (660 μF) as the energy storage part and the commercial thermo-hygrometer (rated working current: 55 μA) as the energy consumption unit, whose circuit is shown in Supplementary Fig. 13. The voltage of capacitor (660 μF) monitored by voltmeter at different MDC-TENG working conditions is shown in Fig. 5e. At initial stage, MDC-TENG is out of work, the capacitor powers the hygrometer alone, resulting in the decrease in capacitor voltage. When the

MDC-TENG begins to work, the voltage of capacitor rises because MDC-TENG provides additional energy, which not only offsets the consumption of hygrometer, but also charges the capacitor. This indicates the excellent output performance of MDC-TENG. However, when the MDC-TENG stops working, the voltage turns to reduce due to the consumption of thermo-hygrometer. Taking advantages of the DC and high output, the MDC-TENG can directly drive the electronic devices (e.g., thermo-hygrometer) without any auxiliary electronic components, as shown in Fig. 5f and Supplementary Movie 2, and the corresponding circuit is shown in Supplementary Fig. 13. The output energy of MDC-TENG can direct drive small electronic devices or charge the energy storage device in a short time, showing its great potential in the application of harvesting mechanical energy. Similar to the traditional sliding AC-TENG, the mechanical wear (the scratches on the PTFE film in Supplementary Fig. 14) is also an inevitable problem for MDC-TENG after long-term working test, but we can overcome this problem through optimizing the device (replacing sliding motion to rolling movement) or introducing liquid lubrication.

## Discussion

In summary, contributed by the microstructural design, we provide an MDC-TENG device with rationally patterned electrode structure, whose triboelectrification charges on the friction charged dielectric surface can be released by electrostatic breakdown and collected by CCEs repeatedly. The effective surface charge density of MDC-TENG (with the size of 1 cm × 5 cm) increases with the electrode structure factor ($k$), reaching 5.4 mC m$^{-2}$ with $k = 50$, which is a milestone of TENGs. More interestingly, the MDC-TENG realizes the miniaturized device structure with high output, and the output characteristic shows good relationship with motion vector parameters (velocity, acceleration, and distance). This provides a huge potential applications in miniaturized electronic device systems as energy supply resource or in MEMS as sensor unit. On the other hand, the charge density can be further improved not only by the finer optimization of device structure and preparation technology via micro/nano processing technology to improve $k$-value furthermore in the future, but also by the enlargement of the DC-TENG size. The latter optimization method can overcome the charge density and current density attenuation of AC-TENG with the device size increasing, which provides a paradigm shift of the large-scale energy-harvesting system for TENGs.

## Methods

**Preparation of MDC-TENG**. An acrylic sheet was cut into rectangle slider (35 mm × 10 mm × 5 mm) as the substrate using laser cutter. Controlling the laser power and laser cutting speed, grooves with different depths are carved alternatively in the middle of the substrate. The copper wire (Φ = 250 μm) and stainless-steel wire (Φ = 100 μm) are used as FE and CCE, respectively. All FEs and CCEs are embedded into the grooves and converged, respectively, into one wire at the bottom of the acrylic substrate for electrical performance. FEs are carefully polished to form height difference between CCEs (gap distance). The friction layer (PTFE) was pasted on the acrylic substrate (200 mm × 20 mm × 5 mm) with a thin layer of foam as the buffer layer to achieve the soft contact between FE and friction layer. Based on the above preparation process, MDC-TENG devices with different number of FEs or with different gap distance were prepared, respectively.

Rotary MDC-TENG was prepared is also prepared by the same method. A cyclic annular acrylic substrate (inside and outside diameters are 70 and 90 mm, respectively) with grooves was cut by laser cutter. About 100 copper FEs (Φ = 250 μm) and stainless-steel CCEs (Φ = 100 μm) are embedded into grooves, respectively. A circular acrylic substrate (Φ = 100 mm) with the PTFE film is used as a rotator.

**Characterization and electrical measurement**. SEM (S4800, Hitachi, Japan) was used to obtain the micrographs of the MDC-TENG device. Linear motor (TSMV120-1S) and commercial motor (80BL165S75-3130TK0) were utilized in the sliding and rotary process measurement, respectively. The short-circuit current and output charges were measured by the programmable electrometer (Keithley Instruments model 6514). The open-circuit voltage was obtained by the mixed domain oscilloscope (MDO3024). The potentiostat (Bio-logic VSP-300) was utilized to monitor the voltage of capacitors during the charging capacitor process and charging/discharging curve of the self-charging power system.

## Data availability
The data that support the findings of this study are available from the corresponding author upon reasonable request.

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

## Acknowledgements

Research was supported by the National Key R&D Project from Minister of Science and Technology (2016YFA0202704), National Natural Science Foundation of China (Grant numbers 61774016, 51432005, 5151101243, and 51561145021), and China Postdoctoral Science Foundation (2019TQ0361, 2019M660587, and 2020M672962).

## Author contributions

Z.Z., Y.D., D.L, J.W., and Z.L.W. conceived the idea, analyzed the data, and wrote the paper. Z.Z., D.L., and J.W. designed the structure of the triboelectric nanogenerators. L.Z. and S.L. helped with the experiments. All the authors discussed the results and commented on the manuscript.

## Competing interests

The authors declare no competing interests.
