## [Peer Review File · Nature Communications]

REVIEWER COMMENTS

Reviewer #1 (Remarks to the Author):

The manuscript "Rationally patterned electrode of direct-current triboelectric nanogenerators for ultrahigh effective surface charge density" proposed microstructure-designed direct-current TENG (MDC-TENG) with rationally patterned electrode structure which can enhance its surface charge density effectively. The authors proposed a noble structural design that can solve the limitation factors of the existing TENG. As much as a new strategy is proposed, mechanisms and parameters to explain this have been proposed, and these have been well explained both theoretically and experimentally. In addition, despite its very small size, there was no difficulty in driving commercial LEDs and electronic devices in real time. I recommend this article for publication in Nature Communications. However, I have some questions which should be addressed before final acceptance.

1. In Fig. 1d and Supplementary Fig. 1, the authors drew a schematic diagram to explain the working mechanism. The working mechanism is well explained, but it will be clearer if you explain the detailed process of where the output starts and ends with the single peak obtained through the experiment.
2. In Figure 3, the authors check the current tendency under different distances, velocities, and accelerations. In the case of distance, the unit time seems to be the same for different distances. In other words, it has the same meaning as Fig. 3c and 3d comparing velocity as a variable. Therefore, Fig. 3a and 3b should be supplementary figure or the meaning of distance as a variable should be further described.
3. In Fig. 4g, the authors compared the charge density according to the gap distance. According to the mechanism that you suggested, when the gap distance becomes 0, the air breakdown does not occur, and the output is expected to decrease significantly. Can you show this experimentally?
4. In Fig. 4h, the authors calculated the electric field strength according to the gap distance through simulation. As the gap distance increases, the absolute value of the electric field also seems to decrease. If it increases beyond a certain level, it seems that the theoretical gap distance in which air breakdown does not occur can be calculated. After this point, the output seems to be constant. Can you show this experimentally?
5. In Figure 5, the authors made a rotating rotary MDC-TENG. In case of general rotating mode TENG, despite friction between the two flat surfaces, mechanical wear is one of the critical issues. I think your MDC-TENG may also have this problem. Can you add SEM images after the experiment along with a long-term stability test?
6. According to what the authors described in figure 3, your output appears to be proportional to speed. In Fig. 5c, as the rotational speed reaches from 120 to 480, the output seems to be proportional to the rotational speed, but only when the rotational speed is 600, the output is increasing rapidly. I hope you can explain this.
7. In Supplementary Fig. 2 the authors check the effects of electrode width on the output performance of DC-TENG. When using copper foil, the contact length and width are the same. However, as the width becomes shorter, it seems that copper wire is used. In this case, the contact length and width seem to be different. It would be appropriate to measure the contact length and not simply indicate the width.

Reviewer #2 (Remarks to the Author):

This is exciting manuscript. This manuscript is based on a previous paper from the author's group. The authors use the principle of the electrostatic breakdown to produce the DC signal. The original electrostatic device output has been enhanced significantly. It is the highest charge density compared all the previous work. It is also important to point out that that device has been tested for different parameters; distance of the electrode, the gap between the device, and friction layer.

Use of simple materials such as, stainless steel, copper, and PTFE to construct the device is a brilliant method. The manuscript is very well written. The figures and data are of very high quality. The interpretation is consistent with their previously published papers.

Is it possible to mention the material type in the figure as well as in the structural design section? Previous paper used copper for FE and CCE. What is the reason for using stainless steel? Does the material of CCE matter?

I understand, in the previous paper the author used copper as friction electrode. However, according to the triboelectrification series other metal may have higher positive charge. Could author provide the reason for using copper?

Do the authors test other materials as friction layers? Will other friction layer provide a better production of charges?

What is M and D means in line 124? It will be better if there is a sentence to explain it.

Compared to the single electrostatic breakdown device, how much does this technique improve?

From the graph, it looks like there are a couple electrostatic breakdown devices that connect to each other. If you use a one single device, what will the performance be?

The authors should include more about the novelty as justification for publication of this manuscript in the Nat. Comm. Isn't this manuscript more of an incremental enhancement over the previously published paper?

I recommend publication of this manuscript after a major revision.

Previous paper relates to this one

<https://advances.sciencemag.org/content/advances/5/4/eaav6437.full.pdf>

Response to Reviewers

The following is a point-to-point response to the reviewer's comments.

Reviewer #1

The manuscript “Rationally patterned electrode of direct-current triboelectric nanogenerators for ultrahigh effective surface charge density” proposed microstructure-designed direct-current TENG (MDC-TENG) with rationally patterned electrode structure which can enhance its surface charge density effectively. The authors proposed a noble structural design that can solve the limitation factors of the existing TENG. As much as a new strategy is proposed, mechanisms and parameters to explain this have been proposed, and these have been well explained both theoretically and experimentally. In addition, despite its very small size, there was no difficulty in driving commercial LEDs and electronic devices in real time. I recommend this article for publication in Nature Communications. However, I have some questions which should be addressed before final acceptance.

1. In Fig. 1d and Supplementary Fig. 1, the authors drew a schematic diagram to explain the working mechanism. The working mechanism is well explained, but it will be clearer if you explain the detailed process of where the output starts and ends with the single peak obtained through the

experiment.

Response: Thank you for your advice. To make the mechanism clearer, we explained the detailed process of DC-TENG at different movement conditions and its corresponding position on the single DC peak. Thus, the Supplementary Fig. 1 was redrawn and the Supplementary Note 1 was revised to provide the detailed working mechanism in the revised manuscript.

Correction:

In Supplementary Materials, the Supplementary Fig. 1 was redrawn as follow:

Supplementary Fig. 1. The mechanism of DC-TENG. (a-c) The schematic diagram of DC-TENG at different movement condition and (d) corresponding output current.

The Supplementary Note 1 was revised in revised Supplementary Materials:

The fundamental mechanism of DC-TENG is on the basis of electrostatic breakdown. The schematic diagram of DC-TENG at different movement condition and their corresponding output current are shown in Supplementary Fig. 1. In the initial state (Supplementary Fig. 1a.), the left edge of friction electrode is coincident with PTFE's. Meanwhile, the friction electrode contacts with PTFE, resulting in the positive charges on the friction electrode and negative charges on the PTFE surface due to the contact electrification. Because of the electret nature feature of PTFE film, the negative charges are held and stayed on the PTFE surface for a long time. Thus, when the friction electrode starts to move forward, air breakdown will occur in the gap between the charge collecting electrode (positive charged) and PTFE surface (negative charged) because of a high electrostatic field inside the gap and a weak dielectric strength of air (3 kV mm^{-1}), resulting in the start of DC

output peak (the poison point a in **Supplementary Fig. 1d**. As shown in **Supplementary Fig. 1b** and the inset schematic, during the friction electrode continues moving forward, the continuous air breakdown will make the electrons **continuously** flow from PTFE to the charge collecting electrode, releasing the contact electrification charges of the surface of the PTFE. Simultaneously, the collected charges flow back to the friction electrode through external circuit due to the potential difference, **resulting in continuously direct current in the circuit (the poison point b in Supplementary Fig. 1d)**. The discharging process will stop when the right edge of friction electrode coincides with PTFE's (**Supplementary Fig. 1c**), **at the same time, the DC output ends (the poison point c in Supplementary Fig. 1d)**. As discussed above, the work mechanism of DC-TENG is the electrostatic breakdown process between charge collecting electrode and friction layer surface. The dielectric strength of friction dielectric film will not affect this process.

2. In Figure 3, the authors check the current tendency under different distances, velocities, and accelerations. In the case of distance, the unit time seems to be the same for different distances. In other words, it has the same meaning as Fig. 3c and 3d comparing velocity as a variable. Therefore, Fig. 3a and 3b should be supplementary figure or the meaning of distance as a variable should be further described.

Response: Thank you for your comment. The output performance (output charges and current) were carried out under different distances, velocities and accelerations in Fig.3. In the initial Fig. 3b, the sliding distance and output current show linear relationship. As your mentioned, this relationship might show the same meaning with Fig. 3d because the different sliding distances spend different time and the unit time seems to be the same for different distance. This is mainly because the setting of experimental velocity is the same for different distances. Considering the non-ideal condition, the velocity is also different for different distances, the output current will not show linear relationship with the various distance. Thus, to rule out the motion time influence on the relationship between MDC-TENG output and various distance, the relationship between output charges and distance was recalculated in the Supplementary Note 3. As shown in Supplementary Note 3 and revised Fig. 3b, the output charges (not the current) shows proportional to the sliding distance with a high linearity of ~0.99.

Correction:

Page 8, paragraph 2, we revised “The output charge at each movement process increases with the extension of sliding distance (**Fig. 3b**), accompanied with the increase of I_{sc} (**Supplementary Fig. 4**). The output charge of MDC-TENG is proportional to the sliding distance with a high linearity of ~ 0.99 (the inset in **Fig. 3b**).” in the revised manuscript.

In the revised manuscript, the Fig. 3b was revised as follow:

Fig. 3 Performance of MDC-TENG under different vector parameters. (a,b) The output charges under different sliding distances (the inset is the relationship between output charge and sliding distance). (c) The charge density and (d) short-circuit current of MDC-TENG with various velocities (the inset is the relationship between I_{sc} and velocity). (e) The charge density and (f) short-circuit current of MDC-TENG with various accelerations (the inset is the relationship between dI/dt and acceleration).

In Supplementary Materials, the Supplementary Note 3 was added.

Supplementary Note 3: Relationship between output charge and sliding distance

According to the mechanism of DC-TENG, during the sliding process, in unit distance, the number of the output charges dQ can be given as follow:

where the q is the collected charge density from electrostatic breakdown, W is the width of slider. Thus, within a certain sliding distance l , the whole output charges Q can be written as follow:

As shown in formula (S2) the output Q is linear to the sliding distance, confirmed by the experimental results in Fig. 3a,3b. (S2)

3. In Fig. 4g, the authors compared the charge density according to the gap distance. According to the mechanism that you suggested, when the gap distance becomes 0, the air breakdown does not occur, and the output is expected to decrease significantly. Can you show this experimentally?

Response: Thank you for your suggestion. The direct current output of MDC-TENG is based on the air breakdown in the tiny gap between charge collecting electrode and friction layer. We prepared a TENG device with the same structure like MDC-TENG, but its gap distance is 0, whose schematic diagram and corresponded output performance is shown in Supplementary Fig. 7.

Supplementary Fig. 7. The output performance of MDC-TENG device with gap distance ~ 0 (MDC-TENG unit = 20, sliding distance = 10 cm, the inset is its schematic diagram): (a) charge density, (b) short-circuit current and (c) open-circuit voltage.

When MDC-TENG (gap distance is 0) slides forward and backward, its output charge is shown in Supplementary Fig. 7a, and as your expectation, the charge density under the sliding distance ~ 10 cm is only $50 \mu\text{C m}^{-2}$, which is much lower than that (2.0 mC m^{-2}) of MDC-TENG (gap distance is $35 \mu\text{m}$). It should be noticed that there is no gap between CCE and friction layer, thus, the mechanism of DC-TENG we mentioned in Figure 1, Supplementary Fig. 1 and Supplementary Note 1 does not

work at this condition (gap = 0). It can be explained by the coplanar-electrode DC-TENG model in the Xu's work recently [R1], because the MDC-TENG with gap distance ~0 shows the same device structure with coplanar-electrode DC-TENG. According to the mechanism of coplanar-electrode DC-TENG, with the coplanar-electrode DC-TENG sliding on the dielectric substrate, the front electrode loses electrons and shows positive charges due to the triboelectrification effect, and the positive charge will transfer from the front electrode to the rear electrode to balance the potential difference. The positively charged rear electrode form a huge electrostatic field at the edge of electrode between negatively charged dielectric layer, resulting in the discharge process between them. These processes (triboelectrification, positive charge transit and discharge) are a completed charge transfer cycle, whose mechanism is shown as Figure R1, and the output of coplanar-electrode DC-TENG is similar to the MDC-TENG with gap distance ~0.

Figure R1. The charge transfer process in the coplanar-electrode DC-TENG. [R1]

[R1] Xu G, Guan D, Yin X, Fu J, Wang J, Zi Y. A coplanar-electrode direct-current triboelectric nanogenerator with facile fabrication and stable output. *EcoMat* 2, e12037 (2020).

Correction:

Page 10, paragraph 2, we added “When the gap decreases to 0, the mechanism of DC-TENG (Supplementary Fig. 1 and Note 1) does not work because there is no gap for the occurrence of air breakdown, and thus the output performance significantly decreases (Supplementary Fig. 7).” in the revised manuscript.

In Supplementary Materials, the Supplementary Fig. 7 was added as follow:

Supplementary Fig. 7. The output performance of MDC-TENG device with gap distance ~ 0 (MDC-TENG unit = 20, sliding distance = 10 cm, the inset is its schematic diagram): (a) charge density, (b) short-circuit current and (c) open-circuit voltage.

4. In Fig. 4h, the authors calculated the electric field strength according to the gap distance through simulation. As the gap distance increases, the absolute value of the electric field also seems to decrease. If it increases beyond a certain level, it seems that the theoretical gap distance in which air breakdown does not occur can be calculated. After this point, the output seems to be constant. Can you show this experimentally?

Response: Thank you for your suggestion. According to the simulation results (Figure R2), when the gap distance is over 200 μm , the electric field strength in the gap is close to a constant ($\sim 6.3 \text{ MV m}^{-1}$). This high simulated electric field strength is larger than the air breakdown field strength, but the air breakdown process is not the unique process in the DC-TENG mechanism. After the air breakdown, the charges must be successfully collected by the CCE, generating current in the external circuit. Therefore, with further increasing the gap distance, the charge collecting process will be affected and the charges cannot be collected by the CCE when the gap is large enough, resulting in no DC output. To prove this experimentally, we prepared the DC-TENGs (single electrostatic breakdown device) with different gap distances (from 40 μm to 550 μm) and tested their output performance, as shown in Supplementary Fig. 5. With the increase of gap distance, the output charge significantly decreases. The DC-TENG with tiny gap distance $\sim 40 \mu\text{m}$ achieves a high output charges $\sim 0.1 \text{ pC}$ (sliding distance: 10 cm), but it reduces to 0.007 pC when the gap distance increases to 450 μm . With the gap distance further increasing to 550 μm , there is no direct current output in the external circuit, showing that no continuous charges caused by the air breakdown could be collected by charge collecting electrodes. Thus, the direct current output is a constant close to zero.

Figure R2. The simulated electric field distribution in the 550 μm from the charge collecting electrode to the charged PTFE film surface.

Correction:

In Supplementary Materials, we added the Supplementary Fig. 5 as follow:

Supplementary Fig. 5 The output of DC-TENG (single electrostatic breakdown device) with gap distance from 40 μm to 550 μm : (a) output charge and (b) short circuit current.

Page 10, paragraph 2, we revised “The fundamental mechanism of MDC-TENG is air breakdown in the gap between CCE and friction layer, and excessive gap distance makes it difficult for CCE to collect charges, and there is no DC output in external circuit (Supplementary Fig. 5).” in the revised manuscript.

5. In Figure 5, the authors made a rotating rotary MDC-TENG. In case of general rotating mode TENG, despite friction between the two flat surfaces, mechanical wear is one of the critical issues. I think your MDC-TENG may also have this problem. Can you add SEM images after the experiment along with a long-term stability test?

Response: Thank you for your advice. The mechanical wear between the friction electrode and friction layer is a common problem for all-type sliding mode TENGs. As one of the sliding mode TENGs, the MDC-TENG also has this problem. The SEM images of the PTFE film before and after long-term working test (5000 cycles) were provided as follow:

Supplementary Fig. 14. The SEM images of PTFE film surface: (a) before and (b) after 5000 cycles working test.

After long-term working test, some scratches can be founded on the PTFE surface due to the friction with friction electrode. For sliding AC-TENG, replacing sliding motion to rolling movement [R1,R2], applying automatic mode transition between contact and non-contact mode [R3,R4], and introducing liquid lubrication [R5] have been reported to reduce mechanical wear behavior, which also provide referential values to reduce the mechanical wear behavior of DC-TENG.

[R1] L. Lin *et al.* Robust triboelectric nanogenerator based on rolling electrification and electrostatic induction at an instantaneous energy conversion efficiency of similar to 55%. *ACS nano* **9**, 922-930(2015).

[R2] H. Yang *et al.* Rolling friction contact-separation mode hybrid triboelectric nanogenerator for mechanical energy harvesting and self-powered multifunctional sensors. *Nano Energy* **47**, 539-546(2018).

[R3] S. Li *et al.* Largely improving the robustness and lifetime of triboelectric nanogenerators through automatic transition between contact and noncontact working states. *ACS nano* **9**, 7479-7487(2015).

[R4] J. Chen, H. Guo, C. Hu, Z. L. Wang. Robust triboelectric nanogenerator achieved by centrifugal force induced automatic working mode transition. *Adv. Energy Mater.* **10** 2000886(2020).

[R5] J. Wu, Y. Xi, Y. Shi. Toward wear-resistive, highly durable and high performance triboelectric nanogenerator through interface liquid lubrication. *Nano Energy* **72**, 104659(2020).

Correction:

Page 13, paragraph 1, we added “Like the traditional sliding AC-TENG, the mechanical wear (the scratches on the PTFE film in **Supplementary Fig. 14**) is also an inevitable problem for MDC-TENG after long-term working test, but we can overcome this problem through optimizing the device (replacing sliding motion to rolling movement) or introducing liquid lubrication.” in the revised manuscript.

In Supplementary Materials, we added the Supplementary Fig. 14 as follow:

Supplementary Fig. 14. The SEM images of PTFE film surface: (a) before and (b) after 5000 cycles working test.

6. According to what the authors described in figure 3, your output appears to be proportional to speed. In Fig. 5c, as the rotational speed reaches from 120 to 480, the output seems to be proportional to the rotational speed, but only when the rotational speed is 600, the output is increasing rapidly. I hope you can explain this.

Response: Thank you for your comment. The basic mechanism of MDC-TENG is the air breakdown in the gap between CCE and friction layer. Thus, the air breakdown process and the triboelectrification process will affect the output of MDC-TENG. In Figure 3 and the calculation in Supplementary Notes, the air breakdown process and the triboelectrification process are almost at the

(where

s_a is the collected charge density and W is the width of slider, Supplementary Note 4). According to this formula, the output I is proportional to the velocity because the s_a is constant due to the same air breakdown and triboelectrification condition. However, for the rotary MDC-TENG device, as the rotational speed increasing to 600, the high speed turntable drives the air in the gap to flow with high speed, resulting in the decrease of air pressure in the gap between CCE and friction layer. In addition, the high speed friction process between friction electrode and friction layer will generate heat and increase the temperature. According to our previous research, the decrease of air pressure and the increase of temperature will make the air breakdown process more easily occur^[R1], and thus the s_a is not a constant anymore. As a result, the relationship between current I and velocity v (

) in Supplementary Note 4 is no longer valid, so the output is not proportional to the rotational speed at high speed zone. Thus, we mentioned these possible effects on the output under the high-speed motion condition.

[R1]Liu, D. *et al.* Hugely enhanced output power of direct-current triboelectric nanogenerators by using electrostatic breakdown effect. *Adv. Mater. Technol.* **5**, 2000289 (2020).

Correction:

In Supplementary Note 4, we revised “Thus, bringing formula (S2) into formula (S3), because the $\hat{\epsilon}$ and W is constant when the sliding speed is not intense (continuous high-speed motion will generate extra heat and rising temperature, which will affect the $\hat{\epsilon}^{11}$),” in the revised Supplementary Materials.

7. In Supplementary Fig. 2 the authors check the effects of electrode width on the output performance of DC-TENG. When using copper foil, the contact length and width are the same. However, as the width becomes shorter, it seems that copper wire is used. In this case, the contact length and width seem to be different. It would be appropriate to measure the contact length and not simply indicate the width.

Response: Thank you for your comment. The output performance of DC-TENG with the electrode width was provided in Supplementary Fig. 2. In this experiment, the electrode width of DC-TENG gradually decreases from 10 mm to 0.25 mm, but their electrode contact length and sliding distance are in common (contact length: 10 mm, sliding distance: 10 cm). To make the Supplementary Fig. 2a clearer and reduce misunderstands of readers, we redrew the Supplementary Fig. 2a and labelled the width and length for all DC-TENGs with various width.

Correction:

In Supplementary Materials, we revised the Supplementary Fig. 2 as follow:

Supplementary Fig. 2. The effect of electrode width on the output performance of DC-TENG. (a) The photographs of DC-TENGs with different friction electrode widths (electrode length: 10 mm). (b, c) SEM images of the surface and cross section of DC-TENG. (d) The output charges and (e) short-circuit currents of DC-TENG with various friction electrode widths (electrode length: 10 mm, sliding distance: 10 cm).

Reviewer #2

This is exciting manuscript. This manuscript is based on a previous paper from the author's group. The authors use the principle of the electrostatic breakdown to produce the DC signal. The original electrostatic device output has been enhanced significantly. It is the highest charge density compared all the previous work. It is also important to point out that that device has been tested for different parameters; distance of the electrode, the gap between the device, and friction layer.

Use of simple materials such as, stainless steel, copper, and PTFE to construct the device is a brilliant method. The manuscript is very well written. The figures and data are of very high quality. The interpretation is consistent with their previously published papers.

1. Is it possible to mention the material type in the figure as well as in the structural design section? Previous paper used copper for FE and CCE. What is the reason for using stainless steel? Does the material of CCE matter?

Response: Thank you for your comment. The material types of friction electrode, charge collecting electrode and friction layer also were mentioned in the structural design section and Fig. 1a. In this work, the stainless steel was utilized as the charge collecting electrode, not the copper in our previous work [R1]. This is not because the material of CCE will affect the performance of the material. The main reason is that the stainless steel wire shows a higher strength than the copper wire with the same diameter. In this work, to achieve the miniaturization of MDC-TENG, the size of CCE is only 100 μm , the thin copper wire always breaks during the preparation of MDC-TENG device due to its weak strength. Generally, according to the mechanism of DC-TENG, the material of CCE should only be conductive. Thus, to make device preparation easier, the stainless steel wire is used as the CCE.

Correction:

Page 5, paragraph 2, we revised “The structure of microstructure-designed DC-TENG (MDC-TENG) with rationally patterned electrode is presented in **Fig. 1a-c**, which possesses multiple fine friction electrodes (FEs, **material: copper wire**) and interlaced charge collecting electrodes (CCEs, **material: stainless steel wire**). All of the individual FEs keep a tiny distance with the adjacent CCEs, and there is a very narrow gap existing between the CCEs and the friction layer (**polytetrafluoroethene, PTFE**), as shown in **Fig. 1b.**” in the revised manuscript.

In **Fig. 1a**, we added the material types of friction electrode, charge collecting electrode and friction layer, which is shown as follow:

Fig. 1 Structural design and working mechanism of MDC-TENG. (a, b) **Structural schematic** and (c) SEM image of MDC-TENG (scale bar: 500 μm). (d) Schematic diagram of DC-TENG working mechanism. The schematic diagram of electrons transition process (e) without and (f) with microstructure optimization. (g) Comparison of the charge density of MDC-TENG with different type TENGs.

2. I understand, in the previous paper the author used copper as friction electrode. However, according to the triboelectrification series other metal may have higher positive charge. Could author provide the reason for using copper?

Response: Thank you for your comment. According to the triboelectrification series, some metals, such as Al and Zn, provide lower work functional than copper, and thus they can lose electrons easier and have higher positive charge during the triboelectrification process. However, the pure zinc is brittle and easy to form dense oxide layer on its surface. Aluminum also has the same problem. Their alloys might overcome these problems, but considering the cost and the fabrication process of MDC-TENG device, the copper wire is the best choice in this work.

3. Do the authors test other materials as friction layers? Will other friction layer provide a better production of charges?

Response: Thank you for your advice. Except for PTFE, some common organic films, such as Kapton (Polyimide), PVDF (Poly(vinylidene fluoride)) and PPS (Polyphenylene sulfide), were utilized as the friction layers to obtain the output performance of MDC-TENG for various friction layers. As shown in Supplementary Fig. 11, it can be seen that the different friction layers show various output performance, indicating the different ability to produce charges for different friction layers. Within the sliding distance of 10 cm (frication area: 10 cm^2), the charge density of MDC-TENG (MDC-TENG units: 20) can be achieved to 1.30, 0.67 and 0.33 mC m^{-2} for Kapton, PVDF and PPS films, respectively, which are lower than that of PTFE, showing the better ability of PTFE film for producing charges.

Correction:

In Supplementary materials, the Supplementary Fig. 11 was added:

Supplementary Fig. 11. The charge density of MDC-TENG with various organic films as friction layer.

Page 11, paragraph 2, we added “Except for PTFE, some common organic films, such as Kapton (Polyimide), PVDF (Poly(vinylidene fluoride)) and PPS (Polyphenylene sulfide), were utilized as the friction layers to obtain the output performance of MDC-TENG. As shown in **Supplementary Fig. 11**, it can be seen that the different friction layers show various output performance, indicating the different ability to produce charges. Within the sliding distance of 10 cm (friction area: 10 cm²), the charge density of MDC-TENG (MDC-TENG units: 20) can be achieved to 1.30, 0.67 and 0.33 mC m⁻² for Kapton, PVDF and PPS films, respectively, which are lower than that of PTFE.” in the revised manuscript.

4. What is M and D means in line 124? It will be better if there is a sentence to explain it.

Response: Thank you for your suggestion. The M and D represents the position on the surface of metal friction electrode and dielectric friction layer, respectively. To make the description clearer and reduce misunderstands, we added the annotations to explain them in the revised manuscript.

Correction:

Page 5, paragraph 3, we revised “When the slide block slides forward and contacts with the friction layer, the electron transition from M1 to D1 occurs (M and D represents the position on the surface of metal friction electrode and dielectric friction layer, respectively), and the energy potential barrier difference between two materials becomes lower (**Fig. 1e2**)...” in the revised manuscript.

5. Compared to the single electrostatic breakdown device, how much does this technique improve?

Response: Thank you for your comment. Taking advantage of this technique to enhance the efficiency of contact electrification, compared to the single electrostatic breakdown device (structure factor $k = 1$), the output performance increases linearly with the increase of structure factor k . The effective surface charge density of MDC-TENG with the size of 1 cm × 5 cm can be improved to 5.4 mC m⁻², which is 50-fold than that of the single electrostatic breakdown device (the charge density ~0.11 mC m⁻², shown in Supplementary Fig. 2).

Correction:

Page 11, paragraph 1, we revised “The charge density of MDC-TENG with $k = 5$ is -0.5 mC m^{-2} , but increases to 5.4 mC m^{-2} with $k = 50$, which is 50-fold that of the single electrostatic breakdown device (the charge density -0.11 mC m^{-2} , as shown in **Supplementary Fig. 2**.” in the revised manuscript.

6. The authors should include more about the novelty as justification for publication of this manuscript in the Nat. Comm. Isn't this manuscript more of an incremental enhancement over the previously published paper?

Response: Thank you for your advice. In this work, we provided a creative way to enhance the performance of TENG by increasing the efficiency of contact electrification through the combination of sliding mode with high efficiency triboelectrification, rationally patterned and microstructural electrode design, and dielectric breakdown effect. Thus, taking advantages of the enhanced the efficiency of contact electrification, a record high effective surface charge density -5.4 mC m^{-2} is achieved, which is over 2 times of the state-of-art of various-type TENGs and over 10-fold higher than that of DC-TENG reported in previously published paper. Following your suggestion, we have added an innovative description in the revised manuscript.

Correction:

In abstract, we revised “Thus, the MDC-TENG achieves a record high charge density of -5.4 mC m^{-2} , which is over 2-fold the state-of-art of AC-TENGs and over 10-fold compared to previous DC-TENGs.” in the revised manuscript.

I recommend publication of this manuscript after a major revision.

Previous paper relates to this one

<https://advances.sciencemag.org/content/advances/5/4/eaav6437.full.pdf>

Response: Thank you for your comment. The previous paper published on Sci. Adv. in 2019 provided a DC-TENG device to generate direct current through electrostatic breakdown strategy. The initial DC-TENG provided a relatively low charge density -0.4 mC m^{-2} compared to the AC-TENG

in the same period (the reported highest value of AC-TENG in 2019 is over 1.2 mC m^{-2}), but in this work, a creative way to enhance the charge density of DC-TENG by increasing the efficiency of contact electrification was provided and the MDC-TENG got a significantly increased charge density $\sim 5.4 \text{ mC m}^{-2}$. In addition, charge density can be further improved with the enlargement of the MDC-TENG size, overcoming the charge density and current density attenuation of AC-TENG with the device size increasing.

We have tried our best to improve the manuscript and made the corresponding revisions in the manuscript. We appreciate for the Editors' and Reviewers' warm work earnestly, and hope that the correction will meet with approval. Once again, thank you very much for your comments and suggestions.